# Candidacidal and Antibiofilm Activity of PS1-3 Peptide against Drug-Resistant *Candida albicans* on Contact Lenses

**DOI:** 10.3390/pharmaceutics14081602

**Published:** 2022-07-31

**Authors:** Jong-Kook Lee, Soyoung Park, Young-Min Kim, Taeuk Guk, Min-Young Lee, Seong-Cheol Park, Jung Ro Lee, Mi-Kyeong Jang

**Affiliations:** 1Department of Chemical Engineering, Sunchon National University, Suncheon 57922, Korea; seal9669@naver.com (J.-K.L.); syp5567@naver.com (S.P.); metingym@scnu.ac.kr (Y.-M.K.); kook8634@naver.com (T.G.); 2Department of Clinical Laboratory Science, Daejeon Health Institute of Technology, Daejeon 34504, Korea; mylee365@empas.com; 3LMO Team, National Institute of Ecology (NIE), Seocheon 33657, Korea

**Keywords:** antifungal activity, antimicrobial peptide, candidacidal activity, biofilm

## Abstract

The recent emergence of antibiotic-resistant fungi has accelerated research on novel antifungal agents. In particular, *Candida albicans* infections are related to biofilm formation on medical devices, such as catheters, stents, and contact lenses, resulting in high morbidity and mortality. In this study, we aimed to elucidate the antifungal and antibiofilm effects of a peptide against drug-resistant *C. albicans*. α-Helical peptides in which the sequence of KWYK was repeated twice and four times, designated peptide series 1 (PS1)-1 and PS1-3, respectively, were generated, and the candidacidal activities of PS1-1, PS1-3, and fluconazole against drug-resistant *C. albicans* cells were assessed. The PS1-3 peptide showed higher killing activity than PS1-1 or fluconazole and acted via a membranolytic mechanism. In addition, the PS1-3 peptide exhibited more potent activity than PS1-1 and fluconazole in terms of fungal biofilm inhibition and reduction at the minimum fungicidal concentration on the contact lens surface. Overall, these findings established PS1-3 as a potential candidacidal agent for applications on contact lenses.

## 1. Introduction

*Candida* species can inhabit various parts of the body, including the gut, mouth, skin, throat, and vagina, and can induce candidiasis [1]. These organisms have evolved through interactions with the host and symbiosis in the body, resulting in a pathogenic nature. Typically, the growth of *C. albicans* cells can be inhibited by the host immune defense system and microbiota [2]. However, the immune system can be affected by changes in the pH and chemical composition of the body, owing to alterations in the microbiota after the use of antibiotics [1,2]. Accordingly, *C. albicans* is able to grow and attach to the host cell surface, causing disease. 

Many people wear contact lenses (CLs) to improve eyesight in an aesthetically pleasing manner. Indeed, more than 140 million people worldwide wear CLs; however, these individuals are more prone to ocular surface problems, such as keratomycosis [3]. Keratomycosis, an ophthalmitis, is the most common CL-related corneal infection caused by an attack of the cornea by *C. albicans* [4,5]. *C. albicans* infections caused by CLs increase inflammation in the eyes, thereby increasing the risk of blindness due to retinal damage, as is observed in individuals with keratomycosis [6]. Most lens cleaning solutions consist of surfactants to remove proteins, fats, disinfectants, and preservatives (thimerosal and chlorobutanol) because CL use is susceptible to microorganisms, such as *C. albicans*, on the ocular surface [7,8]. Drug resistance encountered following low-concentration treatment and infection by strains resistant to high-dose treatment are major problems associated with antimicrobial drug use. Biofilm formation is a drug resistance factor that is caused by reduced drug efficacy. CL wearers can experience vision loss due to the misuse of multipurpose solutions and/or infection with drug-resistant *C. albicans* [7]. 

A previous study reported that the peptide series 1 (PS1) peptide has robust antibacterial activity against membranolytic mechanisms, owing to its repeated sequences of XTryptophan(W)ZX (X: Lysine or Arginine, Z: Leucine, Tyrosine, Valine, or Glycine) [9]. The PS1 analog peptides also have potent antibiofilm activity against drug-resistant bacteria but show low toxicity against mammalian cells [10]. However, the candidacidal effects of carious PS1 analogs have not been studied extensively. This study was conducted with the hypothesis that the peptides could act on the fungal membrane and on the extracellular polymeric substance (EPS) of the fungal biofilm based on the previous study [9], that the antibacterial activity of these peptides was achieved by membranolytic action.

Accordingly, in this study, we used PS1-1 (containing two KWYK sequences) and PS1-3 (containing four KWYK sequences) peptides to compare the effects of the differences in the KWYK peptide length on candidacidal and antibiofilm activity. Overall, our findings provide evidence that PS1 analog peptides can be used as antifungal agents in CL solutions.

## 2. Materials and Methods

### 2.1. Materials

Dulbecco’s phosphate-buffered saline (DPBS), Dulbecco’s modified Eagle’s medium (DMEM), propidium iodide (PI), carboxyfluorescein (FAM) was purchased from Thermo Fisher Scientific (Waltham, MA, USA). *C. albicans* CCARM 14007 was obtained from the Culture Collection of Antimicrobial-resistant Microbes (CCARM) in Korea. It was first isolated in 1999 and found resistant for amphotericin B, flucytisine, and fluconazole. Fluconazole-susceptible *C. albicans* (KCTC 7270, from human skin lesion of erosion interdigitalis, Uruguay) was obtained from the Korean Collection for Type Cultures. 

### 2.2. Synthesis of PS1 Analog Peptides

PS1-1 (Lys-Trp-Tyr-Lys-Lys-Trp-Tyr-Lys [(KWYK)_2_]) and PS1-3 (Lys-Trp-Tyr-Lys-Lys-Trp-Tyr-Lys-Lys-Trp-Tyr-Lys-Lys-Trp-Tyr-Lys [(KWYK)_4_]) peptides were synthesized using 9-fluorrenylmethoxycarbonyl solid-phase methods on a Rink amide 4-methyl benzhydrylamine resin (Novabiochem, Darmstadt, Germany; 0.55 mmol/g). The peptides were synthesized using a Liberty microwave peptide synthesizer (CEM Co., Matthews, NC, USA). To generate N-terminal fluorescently labeled 5-FAM peptides, resin-bound peptides were treated with 20% (*v*/*v*) piperidine in dimethylformamide to remove Fmoc protection groups from the N-terminal amino acids. Crude PS1-1 and PS1-3 peptides were purified using high-performance liquid chromatography on a Jupiter C18 column (250 × 21.2 mm, 15-μm, 300 Å), using an appropriate 0–60% acetonitrile gradient in water with 0.01% trifluoroacetic acid at a flow rate of 10 mL/mL and 40 °C. The molecular masses of the peptides were confirmed using a matrix-assisted laser desorption ionization mass spectrometer (MALDI II; Kratos Analytical, Manchester, UK) [9] at linear and positive ion mode in external calibration with polypeptide mass standard.

### 2.3. Antifungicidal Activity

To evaluate the antifungal activity of the PS1 analog peptides against *C. albicans* and *C. albicans* ATCC 14007, cells were grown overnight and then suspended in yeast extract-peptone-dextrose (YPD) medium. A total of 100 µL of spore suspension (1 × 10^4^ cells/mL) was transferred to 96-well microtiter plates and mixed with 100 µL of two-fold diluted PS1 peptides in 10 mM sodium phosphate (SP) buffer and PBS containing 10% YPD media. After incubation for 24 h at 28 °C, inhibition of fungal growth was evaluated microscopically using an inverted microscope. The turbidity in each well was determined by measuring the absorbance at 570 nm using a SpectraMax M5 microplate reader (Molecular Devices, Sunnyvale, CA, USA). The MFC value was defined as the lowest drug concentration that reduced growth by 100% compared with that of the control [10].

### 2.4. FAM-PS1-1 and FAM-PS1-3 Peptide Binding Affinity

The binding affinity of the PS1-1 and PS1-3 peptides to *C. albicans* was assessed. Cells were grown overnight in YPD medium at 37 °C. Aliquots were suspended in PBS containing 10% YPD medium at 5 × 10^5^ cells/mL. FAM-labeled PS1-1 and PS1-3 peptides at one-half the MFC were added to the mixture and incubated for 1 min. *C. albicans* cells were centrifuged and washed to remove unbound peptides. The fluorescence of the cells was measured using FOBI (Cellgentek, Daejeon, Korea) at different times (2, 4, 6, 8, 10, 15, 20, 25, and 30 min) with a SpectraMas M5 ELISA microplate reader (excitation wavelength: 485 nm, emission wavelength: 538 nm) [11]. 

### 2.5. Assessment of Time Kill-Kinetic Activity Using Fluorescence-Assisted Cell Sorting (FACS) Analysis and Fluorescence Microscopy

*C. albicans* cells were grown to mid-log phase at 28 °C. Aliquots were suspended in 10 mM SP buffer and PBS with 10% YPD medium at a density of 5 × 10^5^ cells/mL. The PS1-1 and PS1-3 peptides were added to the mixture at the MFC and incubated for 2, 4, 6, 8, and 10 min. The cells were centrifuged for 10 min at 400× *g*, washed three times to remove unreacted peptides, and then stained with 1 μg/mL PI (Life Technologies, Carlsbad, CA, USA). Cells were evaluated using FACS (Attune NxT; ThermoFisher, Seoul, Korea) [11]. 

### 2.6. Analysis of Morphological Changes Using SEM

Morphological changes in *C. albicans* CCARM 14007 cells were assessed by comparing cells in the absence or presence of PS1-1 and PS1-3 peptides at the MFC. *C. albicans* ATCC 14007 cells grown to mid-logarithmic phase at 28 °C were resuspended in PBS (pH 7.2) with 10% YPD medium to a concentration of 5 × 10^5^ cells/mL, after which each peptide was added at the MFC. After incubation for 1 h at 37 °C, the cells were fixed (4% paraformaldehyde), dehydrated using a 50–100% ethanol series (10 min at each step), and coated with platinum. The cells were examined by SEM (JSM-7100F; JEOL, Ltd., Tokyo, Japan).

### 2.7. Biofilm Inhibition Assay

Biofilm formation by *C. albicans* CCARM 14007 was assessed in the absence or presence of PS1-1 and PS1-3 using microdilution assays. Briefly, two-fold serial dilutions (1–128 μM) of each peptide were added to YPD medium containing 5 × 10^5^ cells/mL, and the plates were incubated for 48 h at 28 °C with gentle shaking at 50 rpm. The culture medium was discarded by pipetting, the cells were washed three times with PBS, and the biofilms were fixed with absolute methanol for 15 min at room temperature and air-dried. To quantify the extent of biofilm formation, plates were stained by adding 100 μL of 0.1% (*w*/*v*) crystal violet (Sigma-Aldrich, St. Louis, MO, USA) to each well for 10 min at room temperature and rinsed three times with PBS to remove the free stain. The fixed stain was then solubilized in 95% ethanol, and inhibition of biofilm formation was determined using a SpectraMax M5 ELISA microplate reader at an absorbance of 595 nm. To visualize live *C. albicans* ATCC 14007 cells, live cells were stained with SYTO 9 (Molecular Probes, Invitrogen; 5 μM; stock solution: 5 mM in dimethyl sulfoxide (DMSO)), incubated in the dark for 30 min at room temperature, rinsed with PBS to remove free SYTO 9 dye, and observed under a fluorescence microscope (OPTINITY KCS3-160S; Korea Lab Tech, Seongnam-si, Korea) [12].

### 2.8. Biofilm Reduction Assay

Drug-resistant *C. albicans* CCARM 14007 was cultured at 28 °C in an appropriate amount of culture medium. The cells were then transferred to 96-well plates (5 × 10^5^ cells/mL) and incubated for 48 h at 28 °C with gentle shaking at 50 rpm. The culture medium was discarded by pipetting, planktonic cells were removed by rinsing the plate twice with 200 μL PBS (pH 7.2), and two-fold serial dilutions (1–128 μM) of each peptide (100 μL) were added to the wells in duplicate. The plates were incubated at 28 °C for 24 h, and the solutions were discarded by pipetting, washed twice with 200 μL PBS (pH 7.2), and fixed with absolute methanol for 15 min at room temperature. The plates were then allowed to air-dry. To quantify the extent of biofilm formation, plates were stained with 100 μL of 0.1% (*w*/*v*) crystal violet (Sigma-Aldrich, Korea) to each well for 10 min at room temperature and rinsed three times with PBS to remove the free stain. The fixed stain was then solubilized in 95% ethanol, and inhibition of biofilm formation was determined using a SpectraMax M5 ELISA microplate reader at an absorbance of 959 nm. To visualize the antibiofilm activity, fluorescence microscopy analysis was performed. EPS components removed from *C. albicans* CCARM14007 were stained using four fluorescent dyes (100 μL of 0.5 μg/mL DAPI for nucleic acids, 100 μL of 1:500 diluted SYPRO red for proteins, and 100 μL of 50 μg/mL FITC-Con A for carbohydrates) and observed using a fluorescence microscope (OPTINIT KCS3-160S; Korea Lab Tech) [11].

### 2.9. CLSM Analysis

To determine the biofilm inhibition and reduction activity of drug-resistant *C. albiacans* CCARM 14007 in the absence or presence of fluconazole (128 μM), PS1-1 (128 μM), and PS1-3 (16 μM), cells were seeded on coverslips (cell culture-treated; SPL Life Sciences, Pocheon-si, Korea). Fluconazole, PS1-1, and PS1-3 were added to PBS (pH 7.2) with 10% YPD medium containing 5% glucose for planktonic growth and/or biofilm formation (cultured for 48 h at 37 °C) by *C. albicans* CCARM 14007 cells (5 × 10^5^ cells/mL). The plates were then incubated for 24 h at 37 °C with gentle shaking at 50 rpm. The solutions were discarded, washed with PBS (pH 7.2), stained with 5 μM FUN1 dye (Invitrogen, Carlsbad, CA, USA), and incubated at 30 °C in the dark for 30 min. The medium was discarded, washed with PBS, and fixed with 4% glutaraldehyde for 20 min. The samples were analyzed using CLSM (A1R HD 25; Nikon, Tokyo, Japan) [7].

### 2.10. Biofilm Activity Assays on Coverslips

The antibiofilm activities of drug-resistant *C. albicans* CCARM 14007 (5 × 10^5^ cells/mL) were assessed on coverslips (cell culture-treated; SPL Life Sciences) in the absence or presence of fluconazole (128 μM), PS1-1 (128 μM), or PS1-3 (16 μM). Fluconazole, PS1-1, and PS1-3 were added to PBS (pH 7.2) with 10% YPD medium containing 5% glucose for assessment of planktonic growth and/or biofilm formation (cultured for 48 h at 37°C) by *C. albicans* CCARM 14007. Plates were then incubated for 24 h at 37°C with gentle shaking at 50 rpm. The culture medium was discarded, and cells were washed with PBS and fixed with 4% glutaraldehyde for 20 min at room temperature. The samples were then washed with PBS and dehydrated using an OTTS shaper for 10 min. After drying with hexamethyldisilazane buffer and coating with platinum, the samples were analyzed by SEM (JSM-7100F; JEOL Ltd., Tokyo, Japan) [11]. 

### 2.11. Biofilm Activity Assay on Soft CLs (SCLs)

Biofilm formation by drug-resistant *C. albicans* CCARM 14007 was evaluated in the absence or presence of fluconanzole, PS1-1 peptide, and PS1-3 peptide by microscopic analysis and CLSM. Fluconazole (128 μM), PS1-1 (64 μM), and PS1-3 (16 μM) were added to PBS (pH 7.2) with 10% YPD medium containing 5% glucose for analysis of planktonic growth and/or biofilm formation (cultured for 48 h at 37 °C) by *C. albicans* CCARM 14007 cells (5 × 10^5^ cfu/mL) on SCLs (ACUVUE^®^; Johnson & Johnson, Jacksonville, FL, USA). The plates were then incubated for 24 h at 37 °C with gentle shaking at 50 rpm. The culture medium was discarded, and the SCLs were washed three times with PBS. Cells were stained with SYTO 9 (Molecular Probes, Invitrogen; 5 μM; stock solution, 5 mM in DMSO) and incubated at 30 °C in the dark for 30 min. The medium was discarded, and the cells were washed with PBS and fixed with 4% glutaraldehyde for 20 min. The samples were then analyzed using CLSM [7].

## 3. Results

### 3.1. Candidacidal Activity of PS1 Analog Peptides

PS1 peptides with repeated sequences of XWZX were previously found to possess potent antibacterial activity and membranolytic mechanisms [9]. These peptides exhibit high antimicrobial activity at low concentrations by binding to lipid head groups and inserting into the lipid bilayer via their charge and hydrophobicity [12]. Here, the candidacidal activities of PS1-1, PS1-3, and fluconazole against standard and drug-resistant (CCARM 14007) strains of *C. albicans* were investigated by determining the minimum fungicidal concentration (MFC). As shown in Table 1, the MFC value of the PS1-3 peptide was better than that of PS1-1 in phosphate-buffered saline (PBS; pH 7.2), and the growth of *C. albicans* CCARM 14007 was significantly inhibited at low concentrations compared with the standard *C. albicans* strain. By contrast, the 128 μM PS1-1 peptide in the PBS and the conventional drug fluconazole in the 10 mM SP buffer or PBS did not inhibit the growth of *C. albicans* CCARM 14007.

### 3.2. 5’-Fluorescein Phosphoramidite (FAM)-Labeled Peptide Binding Activity

The secondary structures of the PS1-1 and PS1-3 peptides in 30 mM sodium dodecyl sulfate micelles were determined in a previous study using PEP-fold modeling [9]. To ascertain whether the PS1-1 and PS1-3 peptides have membrane-binding capacity, we investigated the effects of the FAM-labeled peptides on binding to *C. albicans*. The FAM-labeled PS1-3 peptide showed a stronger membrane binding affinity than the PS1-1 peptide in the 10 mM SP buffer (pH 7.2), whereas PS1-1 showed the strongest affinity to *C. albicans* in the PBS (pH 7.2; Figure 1A). The binding affinity of FAM-labeled PS1-1 and PS1-3 on the membrane of *C. albicans* was also analyzed at one-half the MFC. The PS1-1 peptide showed faster binding to the *C. albicans* membrane than PS1-3 in the 10 mM SP buffer (Figure 1B). However, PS1-3 had stronger binding affinity than PS1-1 in the PBS (Figure 1C). 

### 3.3. Time Kill-Kinetic Activity

To determine the fungicidal activity of the peptides in SP buffer and PBS, *C. albicans* cells were treated with the two peptides at their MFCs, and the cytolytic activity was assessed using propidium iodide (PI) staining (Figure 2A,B). The PS1-3 peptide strongly disrupted cell membranes and quickly increased candidacidal activity in the 10 mM SP buffer and PBS compared with PS1-1. In particular, over 70% of the *C. albicans* cells died within 2 min of treatment with the PS1-3 peptide, whereas the PS1-1 peptide killed less than 30% of the cells.

### 3.4. Analysis of Morphological Changes through Scanning Electron Microscopy (SEM)

Next, we investigated the mechanisms of action of the PS1-1 and PS1-3 peptides at their MFCs using SEM (Figure 3). PS1-1 and PS1-3 were observed on the cell surface of *C. albicans* CCRAM 14007 in the PBS. Moreover, PS1-1 induced membrane pore formation and promoted the release of substances into the cytosol after binding to the membrane. However, PS1-3 caused intense cell membrane damage, such as a “bee hive shape,” when applied at the MFC. 

### 3.5. Antibiofilm Activity of PS1 Analog Peptides

*C. albicans* forms biofilms on the surface of host cells, thereby enhancing antifungal drug tolerance. Accordingly, biofilm-forming *C. albicans* can cause difficult-to-treat infections. Although long-term studies of antimicrobial agents have been conducted, the relationship between biofilm formation and drug tolerance is not yet clear. Therefore, we next investigated the inhibitory effects of PS1-1 and PS1-3 on biofilms formed by *C. albicans*. The biofilm formation was inhibited in a dose-dependent manner by the PS1-1 and PS1-3 peptides (Figure 4A). Specifically, the 16 μM PS1-3 peptide inhibited the *C. albicans* biofilm formation by up to 95% after 24 h of incubation, whereas the 128 μM PS1-1 peptide inhibited biofilm formation by about 70%. These results suggested that the biofilm inhibitory effects of PS1-3 against *C. albicans* were more potent than those of PS1-1. Furthermore, biofilm formation by *C. albicans* was reduced in a dose-dependent manner by the PS1-1 and PS1-3 peptides (Figure 4B). Notably, the 16 μM PS1-3 peptide reduced the biofilm formation by approximately 50%. The antibiofilm activities of the PS1-1 and PS1-3 peptides were also observed using fluorescence microscopy (Figure 4C). When applied at a concentration of 16 μM, PS1-3 reduced the planktonic cell growth of *C. albicans* to a greater extent than PS1-1 applied at a concentration of 128 μM. Moreover, 16 μM PS1-3 also blocked the secretion of extracellular polymeric substances (EPSs) secreted by *C. albicans*, such as nucleic acids, proteins, and carbohydrates, which stabilize biofilm formation (Figure 4D). 

### 3.6. Ex Vivo Antibiofilm Activities of PS1-1 and PS1-3 Peptides

We next determined whether PS1-1, PS1-3, and fluconazole inhibited the biofilm formation on the cell culture cover glass inoculated with *C. albicans* cells, using confocal laser scanning microscopy (CLSM) and SEM. Treatment with PS1-1 (16 μM) and PS1-3 (64 μM) resulted in significant fluorescence reduction (cell growth inhibition) compared with fluconazole (128 μM; Figure 5, top). The antibiofilm activity of the PS1-3 peptide demonstrated the substantial removal of the biofilm (depth: 66.66 μm) compared with the control (depth: 178.74 μm) in the CLSM analysis, although the activities of PS1-1 (depth: 95.60 μm) and fluconazole (depth: 78.52 μm) could not be clearly observed (Figure 5, bottom). 

The antibiofilm activities of PS1-1, PS1-3, and fluconazole were also monitored using SEM (Figure 6). The PS1-3 peptide showed significant inhibition of the biofilm formation and intense biofilm reduction activity compared with PS1-1 and fluconazole when applied at the MFC.

### 3.7. Antibiofilm Activity on the CLs

To investigate the antibiofilm activity of the peptides on the CL surface, we performed biofilm inhibition and reduction assays using SYTO 9 live-cell dye. We observed that PS1-3 and fluconazole inhibited the biofilm formation by *C. albicans* ATCC 14007 when applied at the MFC (Figure 7A). Interestingly, the PS1-3 peptide had very effective antibiofilm activity (depth: 69 μm) on CLs, whereas fluconazole, an antifungal drug, showed weak antibiofilm activity (depth: 115.32 μm). We also compared the antibiofilm activities of PS1-3 and fluconazole on the CL surface (Figure 7B). Compared with the untreated CL surface (depth: 185.6 μm), the PS1-3-treated CL surface showed substantially reduced *C. albicans* ATCC 14007 biofilm formation. Fluconazole slightly reduced the biofilm formation on the CL surface (depth: 256.16). These data suggested that the PS1-3 peptide had the most potent antibiofilm activities against drug-resistant *C. albicans* ATCC 14007 on CL surfaces. 

## 4. Discussion

CL-associated keratitis/endophthalmitis is mainly treated with antifungal agents [13]. However, *C. albicans* can develop drug resistance to the antifungal components of CL care solutions [7,14]. Keratitis and/or endophthalmitis can occur when CLs covered in biofilms of drug-resistant *C. albicans* are worn. In this study, we provided interesting evidence of the membranolytic mechanism of a novel PS1-related peptide series with a repeated sequence of XWZX [9]. To develop these PS1-1 and PS1-3 peptides as new drugs for the management of keratitis/endophthalmitis, we first evaluated their antifungal activity, binding affinity, and killing kinetics in *C. albicans* planktonic cells. 

The ionic strengths of the PS1-1 and PS1-3 peptides were important for their antifungal activity against *C. albicans*. The KWYW repeat peptide sequence resulted in aggregation in the various buffer solutions. In particular, the peptide showed stronger aggregation in the 10 mM SP buffer (pH 7.2) than in the PBS (pH 7.2). Therefore, we compared the antifungal activities of the PS1-1 and PS 1-3 peptides in the 10 mM SP buffer and PBS. The short PS1-1 peptide sequence had strong antifungal activity, owing to increased hydrophobicity and strong aggregation at low ion concentrations, such as that encountered in the 10 mM SP buffer. However, the activity was low at high ion concentrations (PBS) because the peptide aggregation was resolved. Notably, the antifungal activity of the PS1-3 peptide, which had a longer sequence than PS1-1, was not affected by the buffer conditions (ionic strength). In particular, the peptide showed strong interactions with the *C. albicans* membrane and induced membrane disruption or pore formation, resulting in a “bee hive shape.” Although the PS1-1 peptide was able to attach to the cell membrane, no substantial damage to the cell membrane was observed. 

*C. albicans* is a pathogenic strain that causes various diseases via host infections [15,16,17]. The continuous use of antifungal agents confers resistance to drug efflux, degradation, gene mutation, and biofilm formation [14]. All antifungal agents can cause side effects, and long-term inappropriate use can lead to the development of antibiotic-resistant strains through drug efflux, degradation, gene mutation, and biofilm formation [12]. Biofilm formation often occurs with more than 100-fold antifungal drug resistance compared with planktonic cells [18]. Importantly, the toxicity of *C. albicans* infection is highly related to its biofilm formation capacity, and many clinical isolates of *C. albicans* show resistance to commercial antifungal medications, such as fluconazole [19,20]. In this study, we first examined the antifungal activities of PS1-1, PS1-3, and fluconazole against *C. albicans*. The peptides showed more potent antifungal activity against planktonic and biofilm-forming bacteria than fluconazole. To confirm the reduction in biofilm formation by PS1-1 and PS1-3, we fluorescently labeled EPS components, such as carbohydrates (fluorescein isothiocyanate [FITC]-Con A), proteins (SYPRO-Red), and nucleic acids (DAPI). Our data indicated that PS1-3 blocked the secretion of EPSs at a low concentration (16 μM) compared with 128 μM PS1-1. We also demonstrated the antibiofilm activities of PS1-1, PS1-3, and fluconazole against *C. albicans*. The PS1-3 peptide (16 μM) showed more active biofilm inhibition and reduction than fluconazole (128 μM). 

In this study, we assessed the antibiofilm activities of antifungal peptides against the *C. albicans* biofilm [21,22]. *C. albicans* biofilms can form during CL storage in disinfection solutions [6,17]. The PS1-1 and PS1-3 peptides were used to investigate the growth inhibition of *C. albicans* cells and were compared with the antifungal drug fluconazole. Previous studies have reported that peptides can disrupt the fungal cell membrane by electrostatic interactions with the anion lipid head and hydrophobic interactions with membrane lipids [23,24,25]. The PS1-3 peptide showed initial electrostatic interactions with the anionic bilayers of *C. albicans* and hydrophobic interactions with membrane lipids, owing to the peptide net charge, which was affected by lysine and the hydrophobicity of tryptophan. In particular, the PS1-3 peptide, which contained four repeating KWYK sequences, showed more robust antifungal activity than the PS1-1 peptide, which contained only two repeating KWYK sequences. Therefore, treatment with the PS1-3 peptide resulted in disruption of the *C. albicans* membrane, yielding a “bee hive shape.” When applied at the MFC, the PS1-3 peptide showed stronger antibiofilm activity on the CL surface compared with PS1-1 and fluconazole. Moreover, the PS1-3 peptide showed weak antibiofilm activity on the CL surface after removing the EPSs, such as nucleic acids, proteins, and carbohydrates. 

Although our study has limitations in evaluation for MIC values with one fluconazole-resistant *C. albicans* strain, there is no doubt that antimicrobial peptides can be used as an alternative to antibiotics. As shown in the results of this study, PS peptides can be used as CL preservatives or an antimicrobial agent, or CLs containing antimicrobial activity can be made by PS peptide-grafting onto the CL surface. In addition, in order to commercialize this peptide, it will be necessary to study in vivo toxicity and stability to proteolytic enzymes.

## 5. Conclusions

In summary, we found that the PS1-3 peptide, which contained repeats of the XYZX amino acid sequence, rapidly attached to the *C. albicans* membrane, and its α-helical structure facilitated membrane disruption. This peptide also exhibited potent biofilm inhibition and reduction activity against *C. albicans* CCARM 14007 on the CL surface. This result may be related to the four repeated KWYK sequences compared with the PS1-1 peptide, which contained only two repeats, and the peptide also exhibited higher candidacidal activity and increased capacity for removal of EPSs produced by *C. albicans* biofilm on the CL surface compared with fluconazole. Our findings provide critical insights into the design of antifungal peptides as CL preservatives to prevent *C. albicans* infections, which can cause keratitis/endophthalmitis.

## Figures and Tables

**Figure 1 pharmaceutics-14-01602-f001:**
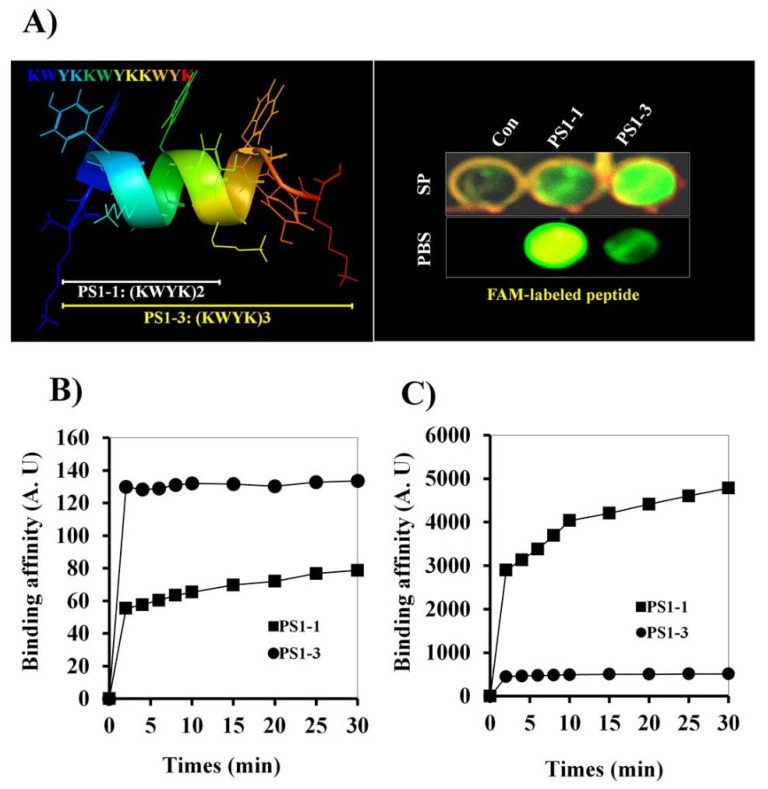
Binding affinity effects of FAM-labeled PS1-1 and PS1-3 peptides against *C. albicans*. PS1-3 peptide structure was modeled using the PEP-FOLD server (http://mobyle.rpbs.univ-paris-diderot.fr/cgi-bin/portal.py#forms::PEP-FOLD, accessed on 10 September 2021) (**A**). At the 1/2 MFC, binding effects of PS1-1 and PS1-3 FAM-labeled peptides on the membrane of *C. albicans* (5 × 10^5^ cells/mL) (**B**,**C**). FAM-labeled PS1-1 and PS1-3 peptides were incubated with *C. albicans* at room temperature (**B**: SP buffer, **C**: PBS buffer). All values represent three individual experiments.

**Figure 2 pharmaceutics-14-01602-f002:**
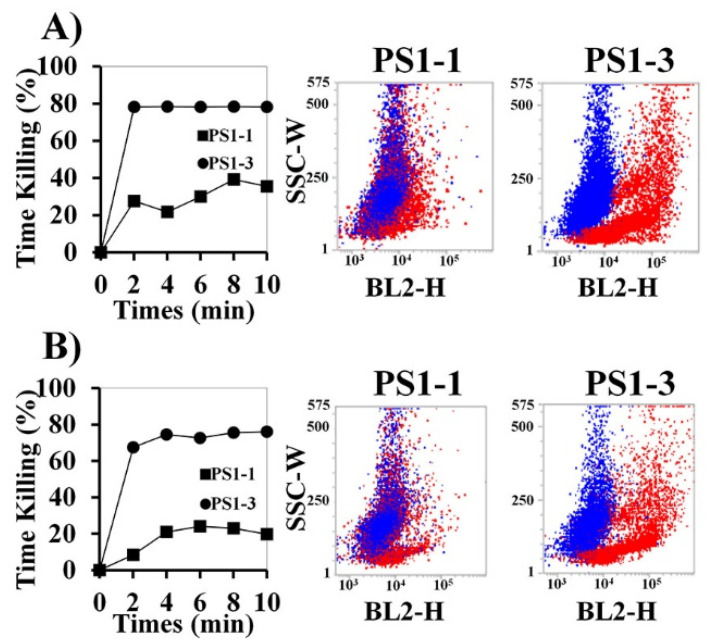
Inhibitory effects of PS1-1 and PS1-3 peptides against *C. albicans* cells. Time-kill kinetics of PS1-1 and PS1-3 peptides with *C. albicans* (5 × 10^5^ cells/mL) within SP (**A**) and PBS buffer (**B**). The cells were stained with propidium iodide (PI) and recorded per each 2 min interval by FACS. Blue and red dots present non-treated and treated cells, respectively. All values represent three individual experiments.

**Figure 3 pharmaceutics-14-01602-f003:**
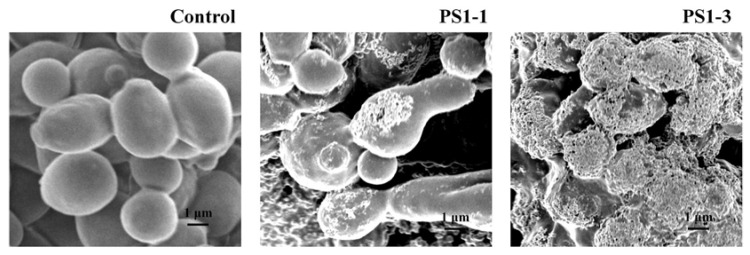
Morphological characterization of cell surface of the peptide-treated *C. albicans* ATCC 14007. After incubation of *C. albicans* CCARM 14007 cells with/without PS1-1 or PS1-3 peptide at the MFC for 1 h, membrane damage values of the cells were examined by using a scanning electron microscope (SEM). Scale bar = 1 μm.

**Figure 4 pharmaceutics-14-01602-f004:**
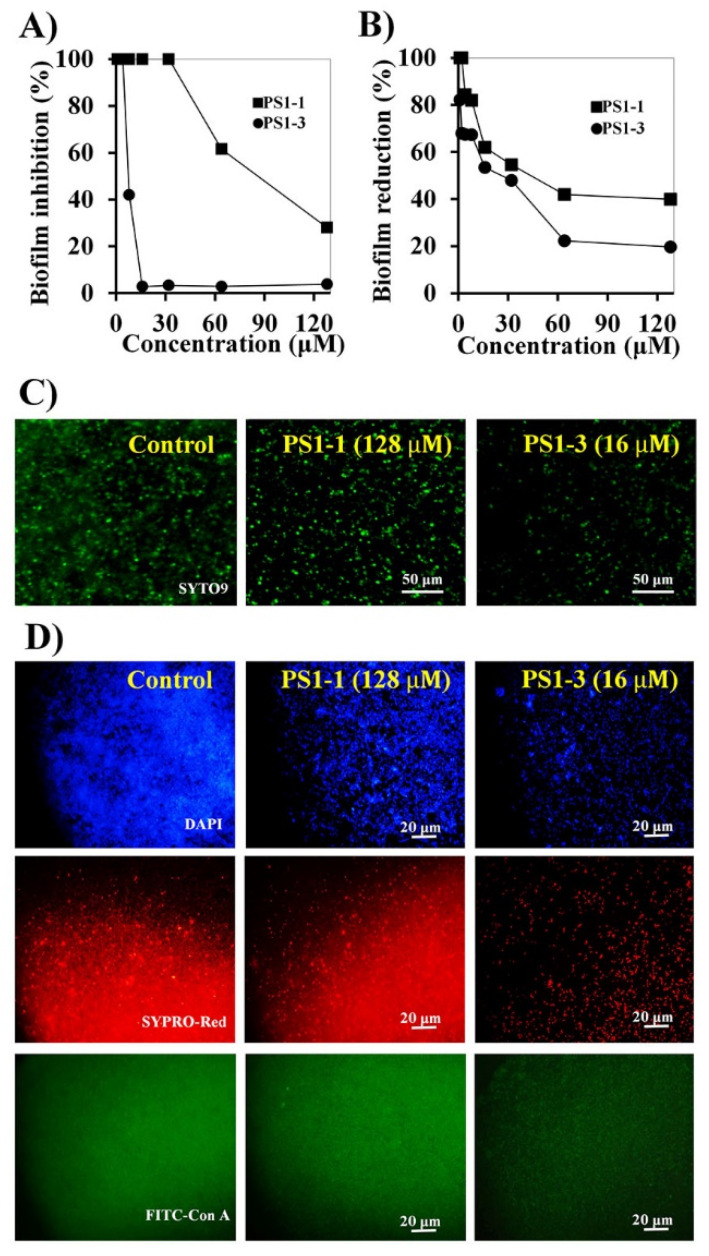
Anti-biofilm activity of PS1-1 and PS1-3. Biofilm inhibition (**A**) and reduction (**B**) in *C. albicans* CCARM 14007 cells strain by treating PS1-1 and PS1-3 peptides. Fluorescence microscope images of PS1-1 (128 μM)- and PS1-3 (16 μM)-treated cells indicated biofilm inhibition (**C**) and biofilm reduction activity (**D**).

**Figure 5 pharmaceutics-14-01602-f005:**
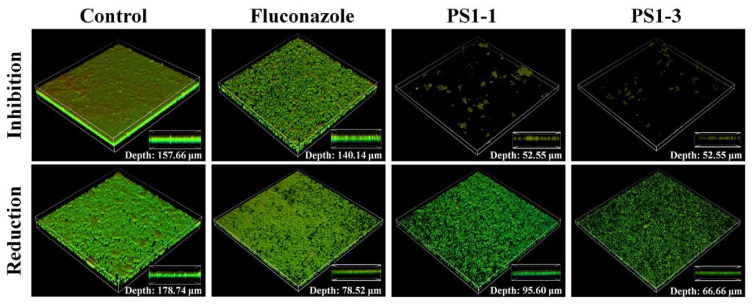
Image analysis of *C. albicans* CCARM 14007 strain using confocal laser scanning microscopy (CLSM). The biofilm inhibition and reduction on *C. albicans* CCARM 14007 in the absence (control) or presence of Fluconazole, PS1-1, and PS1-3. Biofilm formation by *C. albicans* CCARM 14007 was stained using FUN1 dye. All values represent three individual experiments.

**Figure 6 pharmaceutics-14-01602-f006:**
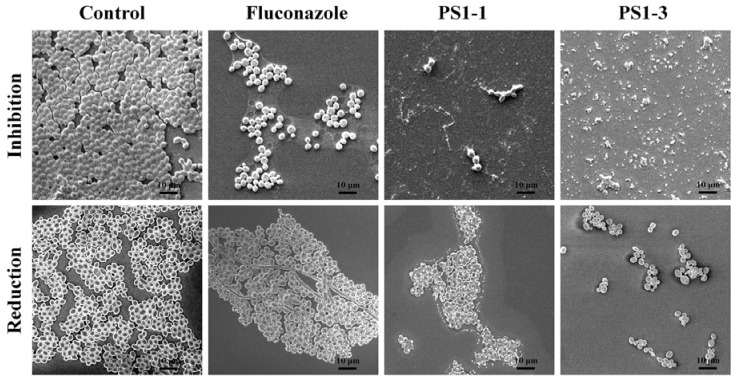
Image analysis of *C. albicans* CCARM 14007 strain using scanning electron microscopy (SEM). The biofilm inhibition and reduction on *C. albicans* CCARM 14007 biofilms in the absence (control) or presence of Fluconazole, PS1-1, and PS1-3 were investigated. Fluconazole, PS1-1, and PS1-3 were treated with MFC for 24 and 48 h. All values represent three individual experiments. The bar is 10 μm.

**Figure 7 pharmaceutics-14-01602-f007:**
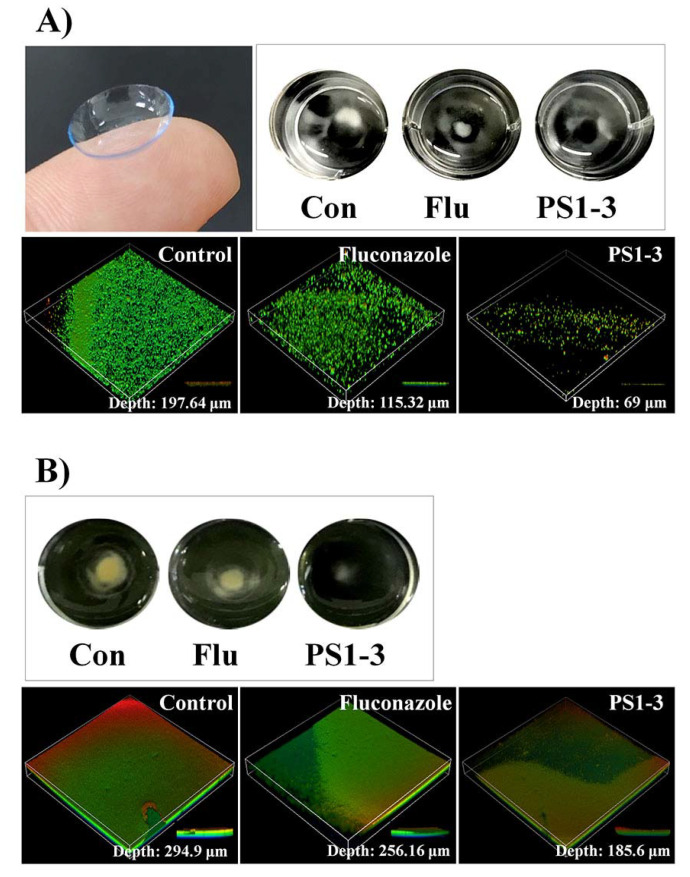
The visual effects of PS1-3 peptide against biofilm on the soft contact lenses (SCL). Biofilm inhibition (**A**) and reduction (**B**) in *C. albicans* CCARM 14007 cells following treatment with Fluconazole (128 μM) and PS1-3 peptide (16 μM).

**Table 1 pharmaceutics-14-01602-t001:** Sequence, molecular weight, and Candidacidal activity of PS1-1, PS1-3, and Fluconazole.

Peptide	Sequence	Molecular Weight (MW)	MIC/MFC (μM)			
			*C. a*		*C. a* (14007)	
			SP	PBS	SP	PBS
PS1-1	(KWYK)_2_	1230 Da	32/32	64/64	32/32	64/128
PS1-3	(KWYK)_4_	2441 Da	8/16	8/16	8/16	8/16
Fluconazole			-	-	>128	>128

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
