# Peer review of "Candidacidal and Antibiofilm Activity of PS1-3 Peptide against Drug-Resistant Candida albicans on Contact Lenses"

_pharmaceutics, 2022, doi:10.3390/pharmaceutics14081602_

Round 1
Reviewer 1 Report
A very interesting article presenting the results of the study at a very high professional level. Taking up the topic is very justified considering the increasing number of mycoses resistant to classical treatment.
I have some comments on the description of the methodology. There is no clarity on the number of Candida albicans strains that were used in the analyses. It seems to me that in the first paragraph it is necessary to precisely list all strains and indicate for which analyzes they were used.
It is also justified to include the limitations of this study in the discussion. First of all, attention should be paid to using only reference strains in the analyses. It is a pity that no clinical isolates were used. Maybe some speculation what results could be obtained for this type of strain. It is known that resistance to fluconazole among Candida is widespread. Would peptides be equally effective for such strains?
I would add a little more to the discussion about the possible clinical aspects of the results of this research. What can be the practical use of peptides and what directions of future research must be taken for implementation?
Author Response
A very interesting article presenting the results of the study at a very high professional level. Taking up the topic is very justified considering the increasing number of mycoses resistant to classical treatment.
â–º We appreciate for your helpful comments, and our manuscript has been revised according to the comments below.
I have some comments on the description of the methodology. There is no clarity on the number of Candida albicans strains that were used in the analyses. It seems to me that in the first paragraph it is necessary to precisely list all strains and indicate for which analyzes they were used.
â–ºWe added information for C. albicans strains used in this study to 2.1 materials.
It is also justified to include the limitations of this study in the discussion. First of all, attention should be paid to using only reference strains in the analyses. It is a pity that no clinical isolates were used. Maybe some speculation what results could be obtained for this type of strain. It is known that resistance to fluconazole among Candida is widespread. Would peptides be equally effective for such strains?
â–º We agree with the reviewer's criticism. However, our laboratory level is not suitable for using clinically isolated strains. In fact, there is a limit to investigate with only strains that can be distributed from national institutions. Although some results not shown in this manuscript, the peptides used in this study exhibited a potent antifungal activity against other strains of fluconazole-resistant Candida (CCARM 14001, 14002, 14004, and 14020). In published many research papers, antimicrobial peptides are suggested as an alternative to antibiotics for fluconazole-resistant Candia because mode of their action is different to conventional antibiotics.
I would add a little more to the discussion about the possible clinical aspects of the results of this research. What can be the practical use of peptides and what directions of future research must be taken for implementation?
â–º Although our study has limitations in evaluation for MIC values with one fluconazole-resistant C. albicans strain, there is no doubt that antimicrobial peptides can be used as an alternative to antibiotics. As shown in the results of this study, PS peptides can be used as CL preservatives or antimicrobial agent, or CL containing antimicrobial activity can be made by PS peptide-grafting onto the CL surface. In addition, in order to commercialize this peptide, it will be necessary to study in vivo toxicity and stability to proteolytic enzymes.
Reviewer 2 Report
I have carefully read the manuscript by Lee et al entitled ‘Candidacidal and antibiofilm activity of PS1-3 peptide against drug-resistant Candida albicans on contact lenses’ in which the authors evaluated the Accordingly, candidacidal and antibiofilm activity of PS1-1 (containing two KWYK sequences) and 56 PS1-3 (containing four KWYK sequences) peptides considering the possible use of PS1 analog peptides can be used as antifungal agents in contact lenses solutions. The study is of interest and well done. The authors perform a plethora of experiments which clearly reveal the potential of the peptides assayed. However, some minor points must be revised before possible consideration for publication in pharmaceutics journal as reported below:
-In the introduction is not clear the origin of these peptides. chemical synthesis? Why the choice of these amino acid motifs?;
-Page 2, line 51: ‘Tyro’ is not an amino acid three letter code. Please correct;
-Page 2, paragraph 2.2. which flow rate for HPLC? Moreover, add more information on the mass spectrometer (e.g. acquisition mode, external calibration and so on)
-Page 2 lines 83-84: indicate the microliter transferred in each well;
-Page 3, lines 103: Which is SP buffer? Authors indicate this at page 5, please correct the mistake, in order to have the full name only the first time it appears, then use abbreviation. For other buffers the full name is not necessary;
-Page 4, lines 192-193: use the full name of aa only the first time it appears, then use abbreviation three letter or mono letter code;
-Table 1. The molecular weight is in Da?.
Author Response
I have carefully read the manuscript by Lee et al entitled ‘Candidacidal and antibiofilm activity of PS1-3 peptide against drug-resistant Candida albicans on contact lenses’ in which the authors evaluated the Accordingly, candidacidal and antibiofilm activity of PS1-1 (containing two KWYK sequences) and 56 PS1-3 (containing four KWYK sequences) peptides considering the possible use of PS1 analog peptides can be used as antifungal agents in contact lenses solutions. The study is of interest and well done. The authors perform a plethora of experiments which clearly reveal the potential of the peptides assayed. However, some minor points must be revised before possible consideration for publication in pharmaceutics journal as reported below:
â–º We appreciate for your constructive and insightful comments, and our manuscript has been revised according to the comments below.
-In the introduction is not clear the origin of these peptides. chemical synthesis? Why the choice of these amino acid motifs ?;
â–º Designation of this motif has been well described in results and discussion section of the previous report (ref. 9). This study was conducted with the hypothesis that the peptides could act on the fungal membrane and on the extracellular polymeric substance (EPS) of the fungal biofilm based on the previous study [9], that the antibacterial activity of these peptides was achieved by membranolytic action.
-Page 2, line 51: ‘Tyro’ is not an amino acid three letter code. Please correct;
â–º We corrected.
-Page 2, paragraph 2.2. which flow rate for HPLC? Moreover, add more information on the mass spectrometer (e.g. acquisition mode, external calibration and so on)
â–º We added a flow rate of HPLC and more information of mass analysis in section 2.2.
-Page 2 lines 83-84: indicate the microliter transferred in each well;
â–º One hundred µL of cell suspension was mixed with 100 µL of peptides.
-Page 3, lines 103: Which is SP buffer? Authors indicate this at page 5, please correct the mistake, in order to have the full name only the first time it appears, then use abbreviation. For other buffers the full name is not necessary;
â–º SP was changed to “sodium phosphate (SP) buffer) in line 85.
-Page 4, lines 192-193: use the full name of aa only the first time it appears, then use abbreviation three letter or mono letter code;
â–º ‘KWYK’ was described in introduction section and the full name of aa in lines 50-51 was corrected.
-Table 1. The molecular weight is in Da?.
â–º We added ‘Da’ in Table 1.